# Circulating Tumor DNA-Based Disease Monitoring of Patients with Locally Advanced Esophageal Cancer

**DOI:** 10.3390/cancers14184417

**Published:** 2022-09-11

**Authors:** Lisa S. M. Hofste, Maartje J. Geerlings, Daniel von Rhein, Sofie H. Tolmeijer, Marjan M. Weiss, Christian Gilissen, Tom Hofste, Linda M. Garms, Marcel J. R. Janssen, Heidi Rütten, Camiel Rosman, Rachel S. van der Post, Bastiaan R. Klarenbeek, Marjolijn J. L. Ligtenberg

**Affiliations:** 1Department of Human Genetics, Radboud University Medical Center, 6525 GA Nijmegen, The Netherlands; 2Department of Medical Oncology, Radboud University Medical Center, 6525 GA Nijmegen, The Netherlands; 3Department of Surgery, Radboud University Medical Center, 6525 GA Nijmegen, The Netherlands; 4Department of Radiology and Nuclear Medicine, Radboud University Medical Center, 6525 GA Nijmegen, The Netherlands; 5Department of Radiation Oncology, Radboud University Medical Center, 6525 GA Nijmegen, The Netherlands; 6Department of Pathology, Radboud University Medical Center, 6525 GA Nijmegen, The Netherlands

**Keywords:** esophageal cancer, circulating tumor DNA, neoadjuvant treatment, liquid biopsies, next-generation sequencing

## Abstract

**Simple Summary:**

The standard of care for patients diagnosed with locally advanced esophageal cancer is neoadjuvant chemoradiotherapy followed by surgery. There is a high clinical need to monitor response to neoadjuvant treatment and recognize patients at risk for recurrence to enable individual treatment strategies. Ultradeep sequencing-based detection of circulating tumor DNA in preoperative plasma of patients with locally advanced esophageal cancer can predict which patients have a high risk of recurrence after neoadjuvant chemoradiotherapy and surgery. Circulating tumor DNA-based prediction of the presence of distant metastasis might eventually be used to reconsider surgery and its associated morbidity in patients with detected circulating tumor DNA or stratify patients for adjuvant treatment.

**Abstract:**

Patients diagnosed with locally advanced esophageal cancer are often treated with neoadjuvant chemoradiotherapy followed by surgery. This study explored whether detection of circulating tumor DNA (ctDNA) in plasma can be used to predict residual disease during treatment. Diagnostic tissue biopsies from patients with esophageal cancer receiving neoadjuvant chemoradiotherapy and surgery were analyzed for tumor-specific mutations. These tumor-informed mutations were used to measure the presence of ctDNA in serially collected plasma samples using hybrid capture-based sequencing. Plasma samples were obtained before chemoradiotherapy, and prior to surgery. The association between ctDNA detection and progression-free and overall survival was measured. Before chemoradiotherapy, ctDNA was detected in 56% (44/78) of patients and detection was associated with tumor stage and volume (*p* = 0.05, Fisher exact and *p* = 0.02, Mann-Whitney, respectively). After chemoradiotherapy, ctDNA was detected in 10% (8/78) of patients. This preoperative detection of ctDNA was independently associated with recurrent disease (hazard ratio 2.8, 95% confidence interval 1.1–6.8, *p* = 0.03, multivariable Cox-regression) and worse overall survival (hazard ratio 2.9, 95% confidence interval 1.2–7.1, *p* = 0.02, multivariable Cox-regression).Ultradeep sequencing-based detection of ctDNA in preoperative plasma of patients with locally advanced esophageal cancer may help to assess which patients have a high risk of recurrence after neoadjuvant chemoradiotherapy and surgery.

## 1. Introduction

The incidence of esophageal cancer is rapidly growing in Western countries. It is a highly lethal malignancy, causing over 500,000 annual deaths worldwide [1]. Only 30% of patients with esophageal cancer qualify for curative treatment. The most common strategy consists of neoadjuvant chemoradiotherapy (CRT) followed by radical surgery of the esophagus and the surrounding lymph nodes [2,3]. Esophagectomy is associated with high morbidity (60%) and mortality (5%) [4,5]. Although several studies demonstrate the absence of residual tumor in the resection specimen after esophagectomy in up to 30% of patients, selection of these ‘complete responders’ that probably do not benefit from surgery remains a difficult challenge with current diagnostics [3]. On the other hand, despite the multimodality treatment approximately 50% of patients will ultimately experience recurrence and might benefit from adjuvant therapy [6]. Accurate biomarkers to monitor response to the neoadjuvant treatment are unavailable, but could possibly contribute to more individual treatment strategies, such as active surveillance in complete responders (omission or postponement of esophagectomy), timely resection in patients with too little benefit from CRT or even cancelation of resection because of high chance of metastatic disease. Furthermore, prediction of patients at high risk of recurrence could select patients for administration of adjuvant treatment after resection.

The presence of circulating tumor DNA (ctDNA) in blood offers a relatively noninvasive and real-time approach for monitoring of disease [7,8]. The use of ctDNA to measure treatment response and prognosis has shown promise in various cancer types [9,10,11]. Yet, the number of studies reporting on ctDNA analysis in esophageal cancer remains limited. Large esophageal cancer cohort studies focus mainly on metastatic disease [12,13,14]. Furthermore, several studies used techniques that lack the sensitivity required for response monitoring in locally advanced patients [15,16,17]. Despite the high clinical need, evidence for the clinical utility of ctDNA measurements in locally advanced esophageal cancer patients remains sparse, as most of these studies represent relatively small and heterogeneous cohorts [18,19,20,21,22,23,24].

We studied the putative role of tumor-informed ctDNA analysis using ultradeep sequencing to monitor disease in a large homogeneous cohort of patients with locally advanced esophageal cancer receiving neoadjuvant CRT and surgery.

## 2. Materials and Methods

### 2.1. Patient and Sample Collection

In this observational study locally advanced esophageal cancer patients (cT2-3 with any N and cT4N0) who underwent surgery in the Radboudumc between July 2017 and April 2020 were consecutively enrolled. Both esophageal adenocarcinoma (EAC) and esophageal squamous cell carcinoma (ESCC) patients were included. All patients received neoadjuvant CRT, consisting of carboplatin and paclitaxel with concurrent 41.4 Gy radiation. Patients did not receive adjuvant chemotherapy as this was not given as the standard of care in the Netherlands during the study period. Longitudinal plasma samples were obtained during treatment. Clinical data, like age, gender, cTNM and American Society of Anesthesiology (ASA) score, were collected. For the estimation of tumor size before treatment, the gross tumor volume (GTV, in cm^3^) was used that was calculated prior to radiotherapy by drawing the primary tumor on each relevant slice of the planning computed tomography (CT) scan [25]. Clinical response measurements were performed by comparing positron emission tomography-CT (PET-CT) scans before and after CRT according to Response Evaluation Criteria in Solid Tumors (RECIST) [26]. Follow-up data were collected during three-monthly visits at the outpatient clinic. Survival data were requested from the Netherlands Cancer Registry (NCR). The study was ethically approved by the Internal Review Board of the Radboudumc (CMO 2017-3192). The study meets the criteria of the code of proper use of human samples of the Netherlands and was conducted in accordance with the Declaration of Helsinki. All patients provided written informed consent.

### 2.2. Tumor Tissue Analysis

For all patients, diagnostic tissue biopsies and resection specimen were obtained and histologically evaluated for differentiation and histological subtype of the tumor by a gastrointestinal pathologist (R.S.v.d.P.). Additionally, resection specimens were analyzed for surgical margins, tumor diameter, pathological response and ypTNM status. Pathological response was classified according to the Mandard tumor regression grading system [27].

For molecular analysis tumor DNA was isolated by microdissection from formalin-fixed paraffin-embedded diagnostic biopsies using the Chelex-100 (Bio-Rad, Hercules, CA, USA) method as previously described [28]. DNA concentrations were measured using the Qubit Broad Range kit (Thermo Fisher, Waltham, MA, USA). Tumor tissue-derived DNA (70 ng) was used for library preparation with a customized single-molecule molecular inversion probe-based next-generation sequencing panel and paired-end sequenced with 2 × 150 cycles on a NextSeq 500 instrument (Illumina, San Diego, CA, USA) [29,30]. The panel, covering 40 kb, consists of fifteen genes (*APC*, *ARID1A*, *BRAF*, *CDKN2A*, *CTNNB1*, *ERBB2*, *FBXW7*, *GNAS*, *KRAS*, *NRAS*, *PIK3CA*, *RNF43*, *SMAD4*, *TGFBR2* and *TP53*) and 56 mononucleotide repeat markers to measure microsatellite instability. This panel was designed to cover regions with a high frequency of somatic mutations in gastrointestinal tumors, with a focus on esophageal and colorectal cancer, based on available literature and relevant databases (Catalogue of Somatic Mutations in Cancer (COSMIC) and The Cancer Genome Atlas (TCGA) available via cBioPortal) [31,32]. Data were bioinformatically analyzed using standard procedures, including bioinformatic exclusion of germline variants using population databases and comparison of variant allele frequencies (VAFs) with estimated tumor cell percentage [28,29,30].

### 2.3. Plasma Analysis

Blood samples were collected in special cell-free DNA (cfDNA) collection tubes (Roche, Basel, Switzerland) and processed within four days using two centrifugation steps: first at 1600× *g* for 10 min to isolate plasma and subsequently at 16,000× *g* for 10 min to remove cellular debris. Plasma was stored at −80 °C until further processing. Isolation of cfDNA from 4 to 15 mL plasma was performed with QIAamp Circulating Nucleic Acid kit (Qiagen, Hilden, Germany). DNA concentrations were measured using the Qubit High Sensitivity kit (Thermo Fisher). cfDNA input ranged from 10–54 ng with a mean of 39 ng. Library preparation was performed with the Twist Library Preparation Kit (Twist, San Francisco, CA, USA) in combination with custom adaptors (IDT, Coralville, IA, USA) with dual index and unique molecular identifiers (UMIs). A hybridization capture was performed on the prepared libraries with a customized probe set (Twist) covering 117 kb, including all regions covered by the tissue panel (Appendix A). Paired-end sequencing was performed on a NovaSeq 6000 instrument (Illumina) using 2 × 150 cycles.

For the analysis BCL files were demultiplexed using bcl2fastq Conversion Software (version 2.20, Illumina). Resulting FASTQ files were subsequently aligned to the hg19 reference genome using Burrow-Wheller Aligner (BWA; version 0.7.8 [33]). Aligned reads were grouped and deduplicated using the read specific UMI information (FGBIO, version 0.8.1). Unique reads that were based on only one UMI read (i.e., singletons) were discarded. Samples were sequenced to a total mean depth of 48,680× and after deduplication and filtering the mean depth was 3949×. To detect small somatic variants, variant calling was performed on the filtered reads using Genomic Analysis ToolKit (GATK) Mutect2 (version 4.1.5.0, Broad Institute, Cambridge, MA, USA) and this was compared to the tumor analysis. Only variants detected in tissue were retrieved from the variants called with Mutect2 in plasma. All tumor-specific variants were also manually checked in Integrative Genomics Viewer (IGV) (version 2.4, University of California, San Diego, CA, USA and Broad Institute, Cambridge, MA, USA). Plasma from 22 healthy donors was used with Mutect2 to create a panel of normals. For technical validation the 22 healthy plasma samples were used for exclusion of platform and panel specific artifacts and estimation of variant specific background noise. Based on an inhouse validation using artificial human control template DNA standards (SeraCare, Milford, MA, USA and Horizon Discovery, Cambridge, UK) an optimum value for the limit of detection (LoD) and threshold were determined for a specificity of 99%. A LoD was calculated for every variant by multiplying the mean variant specific background noise by fifteen. Only variants with at least four mutant molecules and a VAF higher than the LoD were designated as true variants. The height of the VAFs of plasma variants were checked to exclude potential germline variants. Plasma samples with at least one tumor-specific variant detected were called positive for ctDNA. The number of mutant molecules per ml plasma was calculated with the mean mutant VAF, volume of plasma used for isolation and the total number of cfDNA molecules.

### 2.4. Data Evaluation

Differences in tumor characteristics and clinical and pathological measurements were compared using the Fisher exact test for categorical variables and the Mann-Whitney (rank sum) test or Kruskal-Wallis test for continuous variables. Correlation was assessed using Spearman rank correlation coefficient. For survival estimates we employed the Kaplan-Meier method. Differences in survival were compared with the Cox Proportional Hazards regression model to estimate regression parameters and hazard ratios (HR) with confidence intervals (CI). Statistical tests were performed in IBM SPSS Statistics (version 25) and figures were generated using R software (version 4.1.2) and GraphPad Prism (version 5.03). All *p*-values were based on two-sided testing and *p*-values < 0.05 were considered significant.

## 3. Results

### 3.1. Patient Characteristics

Diagnostic tissue biopsies of 88 patients with locally advanced esophageal cancer, for whom plasma samples prior to treatment (T0) and prior to surgery (T2) were available, were analyzed for somatic mutations (Figure 1a). With a panel of 15 genes, a total of 132 tumor-specific mutations were detected in 78 of 88 tumors with on average two mutations per tumor. Most mutations were found in *TP53* (60%) and nine patients (12%) had a microsatellite instable tumor (Appendix A).

The 78 patients with tumor-specific mutations were included for ctDNA evaluation. The median age of these patients was 67 years, and the majority of patients were male (77%) (Table 1). All patients completed neoadjuvant CRT and underwent esophagectomy. The majority of patients presented with suspected lymph node metastases (cN+) (65%). A total of 68 patients presented with EAC (87%) and seven patients presented with ESCC (9%). In total, 329 tissue and plasma samples were analyzed. Plasma samples were obtained prior to treatment (T0), one or two weeks after initiation of CRT (on average eleven days after initiation) (T1) and at day of surgical treatment (on average 113 days after initiation of CRT) (T2) (Figure 1b).

### 3.2. ctDNA Detection Pre-CRT Was Associated with Tumor Burden

At least one tumor-specific mutation identified in the paired tissue analysis could be detected in plasma pre-CRT (T0) in 56% (44/78) of patients ( Appendix A). In these patients with detected ctDNA, 74% (64/87) of tumor-informed mutations were detected. ctDNA was detected more often in patients with a more advanced tumor stage (81% in IIIB vs. 17% in IB, *p* = 0.05, Fisher exact) and patients with detected ctDNA had a significantly higher tumor volume (median 54 cm^3^ vs. 32 cm^3^, *p* = 0.02, Mann-Whitney) (Table 1). Absolute ctDNA levels (mean mutant molecules per ml plasma) were associated with tumor volume (ρ = 0.31, *p* = 0.01, Spearman rank) and were significantly higher in patients with cN2 disease compared to cN0 (*p* = 0.02, Kruskal-Wallis) and patients with ESCC compared to EAC, although ESCC numbers were small (*p* = 0.03, Mann-Whitney) (Figure 2).

### 3.3. Preoperative ctDNA Detection Was Associated with Higher Risk of Recurrence

After neoadjuvant CRT, ctDNA was detected preoperatively (T2) in 18% (8/44) of patients with, and in 0% (0/34) of patients without detected ctDNA pre-CRT (T0) (Figure 3a). In patients with detected ctDNA, 100% (13/13) of tumor-informed mutations were detected. No association was found between detection and levels of ctDNA at T2 and pretreatment tumor characteristics (Table 1), clinical response measurements based on PET-CT scans prior to surgery and pathological response evaluation from resection specimen. However, patients with detected ctDNA at T2 had a significantly higher risk of disease progression (HR 2.6; 95% CI 1.1–6.3, *p* = 0.04, Cox-regression) and disease-specific death (HR 3.1, 95% CI 1.3–7.6, *p* = 0.01, Cox-regression) compared to patients with undetected ctDNA levels at T2, independent of their ctDNA status at T0 (Figure 3b,c and Appendix A). No difference was found in progression-free and overall survival for EAC and ESCC patients. Postoperative lymph node status was associated with a higher risk of disease progression and disease-specific death but detected ctDNA at T2 was shown to be independently associated with worse progression-free and overall survival in a multivariable Cox-regression analysis (Appendix A). ctDNA detection at T2 was predictive of development of distant metastasis, which is exemplified by patient 120 who had detected ctDNA at T2, had a complete response in the resection specimen including negative postoperative lymph node status, but was diagnosed with liver metastases already three months after surgery.

### 3.4. ctDNA Dynamics during CRT Was Not Associated with Response

For 37 patients, which were enriched for those with detected ctDNA pre-CRT (T0), a plasma sample collected one or two weeks after initiation of CRT (T1) was assessed and compared to the clinical response measured before surgery and the pathological response measured in the resection specimen from surgery approximately fourteen weeks later. Neither the detection rate nor the levels of ctDNA differed among the different clinical and pathological response groups (Appendix A).

## 4. Discussion

In this study, we examined the clinical validity of ctDNA measurements to monitor response and predict outcome in patients with locally advanced esophageal cancer that underwent neoadjuvant CRT before surgery. With our sensitive sequencing approach ctDNA was detected before neoadjuvant CRT in 56% of patients. After CRT ctDNA was still detected in 10% of patients. This preoperative detection of ctDNA was independently associated with disease progression (*p* = 0.03) and worse survival (*p* = 0.02).

As new treatment strategies are arising for patients with locally advanced esophageal cancer, monitoring of response to neoadjuvant treatment and recognition of patients at risk for recurrence need to be optimized to enable personalized treatment choices [34]. Few studies on ctDNA measurements in the neoadjuvant setting of esophageal cancer patients have been reported [20,21,22,23]. Ococks et al. mainly focused on minimal residual disease detection at the postoperative timepoint and did not use their post-CRT timepoint to correlate ctDNA detection with recurrence [21]. Azad et al. specifically measured ctDNA after neoadjuvant CRT before surgery in only 23 patients [20]. To our knowledge, our work represents the largest cohort to investigate the prognostic value of ctDNA detection after CRT in patients with locally advanced esophageal cancer and comprises a complete set of tissue and plasma samples combined with clinical and pathological parameters.

Prior to treatment we were able to detect ctDNA in 56% of patients with locally advanced esophageal cancer. This percentage is consistent with other studies and confirms that ctDNA detection in these patients is challenging [21,35]. This is in line with the finding that localized esophageal tumors shed low levels of ctDNA in comparison to other localized tumor types and that adenocarcinomas shed lower levels of ctDNA than squamous cell carcinomas [20,36]. As such, in only a subset of patients the levels of ctDNA can be monitored, indicating that ctDNA evaluations are not informative for every patient and thus need to be combined with other modalities, like PET-CT measurements. Furthermore, it also implies that ultrasensitive approaches, like we used here, are required for ctDNA analysis in patients with locally advanced esophageal cancer.

In a multivariable analysis preoperative ctDNA detection was independently associated with disease recurrence and worse survival. Therefore, ctDNA detection may predict outcome prior to surgical intervention and has additive value to current detection methods using PET-CT. Detection of ctDNA was also observed in patients with a complete pathological response in the resection specimen that later developed distant metastases. This indicates that detection of ctDNA at this time point is probably predictive for the presence of occult distant metastasis. This is in line with previous literature describing that ctDNA detection after CRT is stronger associated with distant metastasis than with local recurrence [20,37]. When this observation is confirmed in other esophageal cancer studies, ctDNA-based prediction of the presence of distant metastasis might eventually be used to reconsider surgery and its associated morbidity in patients with detected ctDNA or stratify adjuvant treatment.

ctDNA dynamics one or two weeks after initiation of CRT were not informative for clinical and pathological response to treatment that was evaluated a few months later. Two other studies performed serial ctDNA analyses during CRT in small cohorts of esophageal cancer patients and found an association with response when measuring ctDNA levels after approximately four weeks of neoadjuvant treatment [35,38]. However, as the neoadjuvant treatment strategy in our cohort lasts only five weeks, we wanted to select an earlier timepoint where treatment adaptation, such as cancelation of CRT, would still be possible. Future large studies will have to explore the usefulness of early response monitoring at different time points during neoadjuvant treatment in locally advanced esophageal cancer patients [39].

A limitation of our study is that, although we used tumor-informed mutation analysis with ultradeep sequencing to improve the sensitivity, ctDNA was still only detected prior to treatment in approximately half of the patients. Usage of a larger gene panel could have improved capturing the heterogeneous mutational landscape of esophageal cancers [23,40], thereby increasing the chance that more positions in the genome could be interrogated in the ctDNA which could increase sensitivity of the analysis. Copy number alterations are frequently found in esophageal tumors but the tumor load of patients in this cohort was too low for reliable copy number analysis in ctDNA [32,41]. For ctDNA evaluations to become useful for all patients with locally advanced esophageal cancer perhaps a combinatorial approach of different ctDNA detection options, like fragmentation and methylation analysis, should be evaluated. Another limitation of our study is the collection of only one sample during the neoadjuvant CRT. Additional timepoints were difficult to arrange for patients receiving CRT in external hospitals. The timepoint of the sample that was collected during CRT may have been too early to monitor treatment response. Moreover, the study comprises a small number of patients with squamous cell carcinoma. We consecutively included esophageal carcinoma patients irrespective of histological subtype as they all receive the same treatment. Importantly, no differences were found in progression-free survival and overall survival for the two subtypes.

Our results show that ctDNA detection during or after CRT using ultradeep sequencing in locally advanced esophageal cancer patients is not yet sufficiently sensitive to stratify complete responders for active surveillance instead of surgery. However, preoperative ctDNA detection may help to predict which patients have a high risk of recurrence and may benefit from adjuvant treatment. Interestingly, Kelly et al. recently reported disease-free survival benefit of adjuvant nivolumab administration in esophageal cancer patients with pathological residual disease after neoadjuvant CRT and surgery [34]. However, our results suggest that adding ctDNA detection might identify patients with occult distant metastases despite a local complete pathological response, who could also benefit from adjuvant nivolumab treatment.

## 5. Conclusions

Tumor-informed ctDNA analysis in preoperative plasma of patients with locally advanced esophageal cancer could be a promising tool to predict which patients have a high risk of recurrence after CRT and surgery and can complement current modalities. These findings need validation in larger prospective studies to establish the clinical utility of ctDNA measurements to guide treatment decisions in patients with locally advanced esophageal cancer.

## Figures and Tables

**Figure 1 cancers-14-04417-f001:**
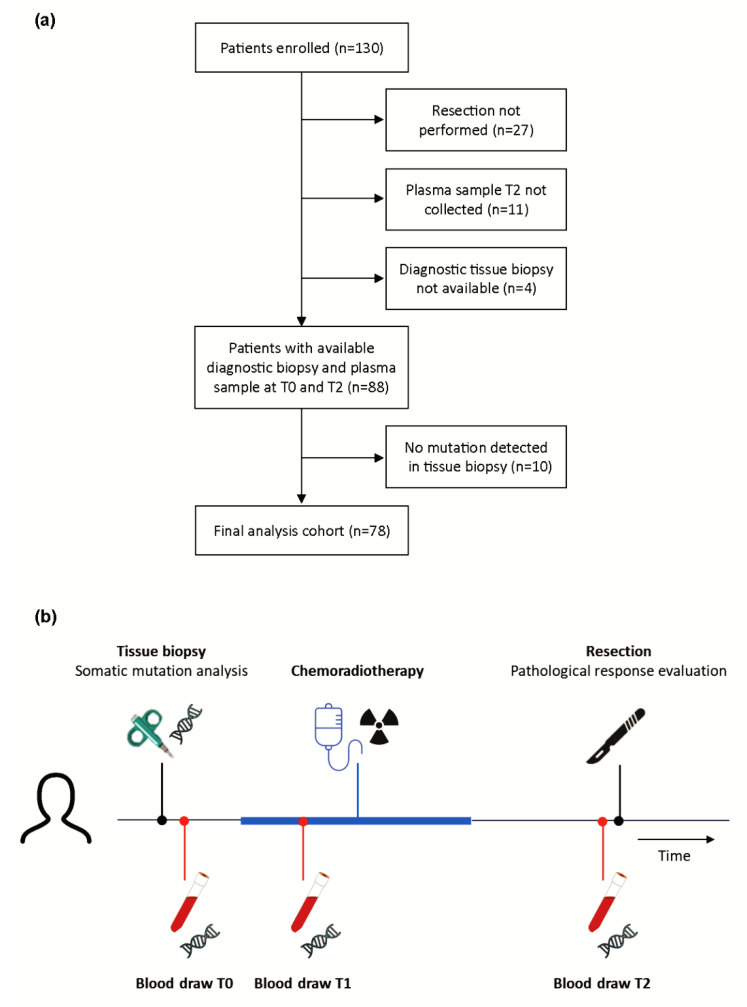
Study design: (**a**) flow chart of patient inclusion; and (**b**) timeline of the study scheme with timepoints of plasma collection pre-CRT (T0), during CRT (T1) and preoperative (T2).

**Figure 2 cancers-14-04417-f002:**
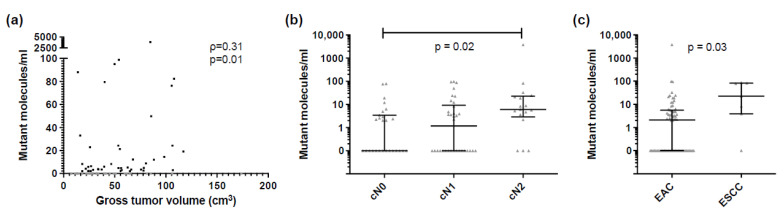
ctDNA levels pre-CRT (T0) and association with clinicopathological characteristics (**a**) Comparison of number of mutant molecules per ml plasma with gross tumor volume in cm^3^. Gross tumor volume of ctDNA negative samples is displayed in grey. Correlation was assessed using Spearman rank correlation coefficient. Comparison of number of mutant molecules per ml plasma with (**b**) suspected lymph node status and (**c**) pathological tumor subtype. Median and interquartile range are shown. *p*-values were calculated using Kruskal-Wallis test and Mann-Whitney test. Esophageal adenocarcinoma (EAC), esophageal squamous cell carcinoma (ESCC).

**Figure 3 cancers-14-04417-f003:**
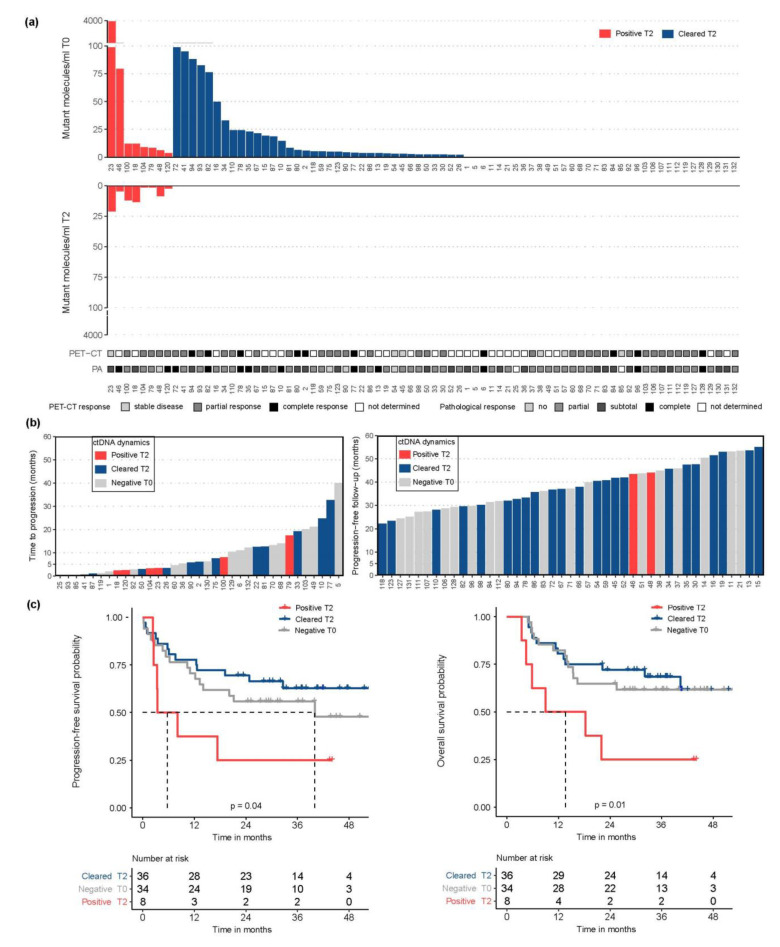
ctDNA dynamics and survival analysis: (**a**) ctDNA levels pre-CRT (T0) and preoperative (T2) combined with PET-CT response measurements and pathological response evaluation in grey. Patients are sorted according to ctDNA levels at T0 and presence of ctDNA at T2; (**b**) the time to progression (left panel) and progression-free follow-up (right panel); and (**c**) Kaplan-Meier analysis of progression-free survival and overall survival dividing patients according to ctDNA dynamics. Red ctDNA detected T0 and T2 (=positive T2); blue ctDNA detected T0, but not T2 (=cleared T2); grey ctDNA not detected T0 and T2 (=negative T0). Positron emission tomography-computed tomography (PET-CT), circulating tumor DNA (ctDNA).

**Table 1 cancers-14-04417-t001:** Clinicopathological characteristics of patients with locally advanced esophageal cancer (*n* = 78). The total number of patients with plasma analysis and the ctDNA detection at T0 and T2 is shown with a percentage. *p*-values were calculated using Fisher exact test and Mann-Whitney test. American Society of Anesthesiology (ASA) score, circulating tumor DNA (ctDNA), chemoradiotherapy (CRT).

Clinicopathological Characteristics		All Patients (*n* = 78)	ctDNA Pre-CRT (T0) (*n* = 78)			ctDNA Preoperative (T2) (*n* = 78)		
			Negative	Positive	*p*-Value	Negative	Positive	*p*-Value
Age, median (range)		67 (50–82)	68 (50–82)	65 (52–80)	0.27	67 (50–82)	64 (54–80)	0.42
Gender, n (%)	Male	60 (77)	24 (40)	36 (60)	0.29	55 (92)	5 (8)	0.38
	Female	18 (23)	10 (56)	8 (44)		15 (83)	3 (17)	
ASA score, n (%)	I	9 (12)	5 (56)	4 (44)	0.76	9 (100)	0 (0)	0.65
	II	46 (59)	19 (41)	27 (59)		40 (87)	6 (13)	
	III	23 (30)	10 (44)	13 (57)		21 (91)	2 (9)	
cTNM stage (7th edition), n (%)	IB (cT2N0)	6 (8)	5 (83)	1 (17)	0.05	6 (100)	0 (0)	0.38
	IIA (cT3N0)	20 (26)	9 (45)	11 (55)		18 (90)	2 (10)	
	IIB (cT2N1)	7 (9)	2 (29)	5 (71)		7 (100)	0 (0)	
	IIIA (cT2N2, cT3N1 or cT4N0)	29 (37)	15 (52)	14 (48)		27 (93)	2 (7)	
	IIIB (cT3N2)	16 (21)	3 (19)	13 (81)		12 (75)	4 (25)	
Gross tumor volume (cm^3^), median (range)		41 (7–174)	32 (7–174)	54 (14–117)	0.02	39 (7–174)	72 (40–88)	0.04
Subtype, n (%)	Adenocarcinoma	68 (87)	32 (47)	36 (53)	0.29	62 (91)	6 (9)	0.40
	Squamous cell carcinoma	7 (9)	1 (14)	6 (86)		5 (71)	2 (29)	
	Adenosquamous carcinoma	2 (3)	1 (50)	1 (50)		2 (100)	0 (0)	
	Undifferentiated	1 (1)	0 (0)	1 (100)		1 (100)	0 (0)	
Differentiation, n (%)	Good-moderate	38 (49)	15 (40)	23 (61)	0.06	35 (92)	3 (8)	0.48
	Poor	35 (45)	19 (54)	16 (46)		31 (89)	4 (11)	
	Not determined	5 (6)	0 (0)	5 (100)		4 (80)	1 (20)	
Follow-up progression free survival (months), median (range)		28 (1–55)	26 (1–53)	30 (1–55)	0.55	29 (1–55)	6 (2–44)	0.12
Follow-up overall survival (months), median (range)		30 (3–55)	29 (5–54)	32 (3–55)	0.90	32 (5–55)	14 (3–44)	0.07

## Data Availability

The data that supports the findings of this study are available within the paper and its supplementary information files or upon request.

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
