# Peer review of "Circulating Tumor DNA-Based Disease Monitoring of Patients with Locally Advanced Esophageal Cancer"

_cancers, 2022, doi:10.3390/cancers14184417_

Round 1
Reviewer 1 Report
The authors explored the use of ctDNA detection prior to chemoradiotherapy (CRT) then prior to surgery to predict residual disease in 88 patients with locally advanced esophageal cancer undergoing standard of care neoadjuvant CRT followed by surgery. The manuscript is generally well-written and structured. The results add to the body of evidence for potential clinically relevant uses of ctDNA in improving disease management of esophageal cancer.
Major comments
1. How exactly is the analysis “tumor-informed”? It is not clear how the use of different panels for the tumor tissue and plasma samples is being analyzed. Are the results from the T0 and T2 plasma restricted to only the regions that overlap with the tissue panel?
2. The lack of matched normal sample sequencing raises concerns about difficulties in distinguishing tumor-derived variants from variants of germline or clonal hematopoietic origin in the ctDNA sequencing. TP53 is a common clonal hematopoiesis gene and that is also the most frequently mutated gene in this study. Additional details on the plasma analysis in the Methods section should be added to address this.
3. Did the authors compare the T2 to the T0 plasma variants (in a “tumor naïve” approach)? It seems the plasma panel covers quite a few additional genes, based on Supplemental Table 1, so that may add additional insights.
4. Please add reference Liu et al. (Froniers in Oncology 2021; https://doi.org/10.3389/fonc.2021.616209) in the Discussion. (Disclaimer: I am not affiliated in any way to that publication; just found it through a quick literature search and it seems relevant to compare, similar to the Ococks et al. and Azad et al. studies).
Minor comments
1. Line 109: Please add more details about the tissue panel, especially how large is it (how many kb).
2. It is not clear what is the performance of the plasma sequencing assay. For example, in Line 146, why was a factor of fifteen was used? Why was a threshold of four mutant molecules used? Can the authors include or cite any validation data with samples with known variants (maybe cell line experiments) that provide more details on typical LOD values of this assay as well as expected sensitivity or specificity?
3. Was a panel of normals created and used with Mutect2?
4. Was MSI calling also performed on the plasma (since the same regions as the tissue panel were included)?
5. Line 168: The authors write that most mutations were found in TP53. What was the exact %?
6. Line 180: For the T2 time point, what was the average number of days after treatment initiation?
7. What was the average number of plasma mutations detected in T0 and T2?
8. A comparison of the biopsy and resection tissue mutations detected might also be interesting to include in the supplement.
9. Supplemental table 2. Please be clear if columns such as Column O (LOD background plasma), Column Q (VAF with Mutect2 T0) and others are frequencies or are percentages (%).
Reviewer 2 Report
Summary:
This manuscript describes ctDNA in EAC and ESCC cases before and after standard neoadjuvant chemoradiotherapy. It aims to highlight the potential of ctDNA to predict recurrence after surgery. The strength of this study is the homogeneous treatment of patients.
Abstract:
ctDNA detection rates pre-treatment (at T0) were split into stage I/II and stage III patients, but the post-treatment value was reported for the combined cohort. Were there stage specific differences post-treatment? In the results section the number of ctDNA positive patients was not reported per stage but across the whole cohort (44/78 patients were ctDNA positive at T0). Is this intentional or could this reporting be consistent?
Methods (Patient and sample collection):
Please indicate that both EAC and ESCC cases were included and analysed. EAC and ESCC are genetically very different. Combining the two subtypes in a genomics study is suboptimal and any outcome conclusions should be made for each subtype separately. I strongly recommend to remove any non-EAC patients from the cohort and repeat the analysis to have a homogeneous cohort.
Results:
- Patient characteristics: Given OAC is a highly mutated cancer type, it might be good to reiterate here that only 15 genes were assessed for mutations to justify the low number of observed mutations. Furthermore, please specify in the text how many EAC and ESCC cases were analysed.
- Table 1: Adding the median follow-up for overall and progression-free survival would be beneficial to describe the patient cohort.
- Supplementary figure 1: Please add patient IDs for easy identification of tumours that showed mutations. I also suggest instead of having a faint line of grey (?) for the ctDNA levels of patients being ctDNA negative to completely remove the line for these samples so the readers can quickly identify ctDNA positive and negative patients. Otherwise, there is only little difference between low levels of ctDNA and no ctDNA.
- ctDNA detection pre-CRT was associated with tumor burden: Comparing 7 ESCC to 68 EAC cases seems too imbalanced to have good statistical power to make a conclusion about ctDNA level differences between the two subtypes.
- Figure 2: This figure is very pixeled in the pdf version I reviewed. Can this be exported in a higher quality before publication?
- Figure 3a: the shades of grey for PET-CT and PA are difficult to separate. especially when the very light-grey boxes are next to white ones (no data available?).
- Figure 3b: Both panels show similar data (time to progression), can both panels be boxplots or boxplots with arrows instead of being a mix? It is a bit confusing why the visualisation changes. Unless the two panels can be combined to one and then the arrows show the patients that haven’t progressed.
Discussion:
- More ctDNA studies in esophageal cancer including samples treated with neoadjuvant therapy have been published recently (e.g. Cabalag et al., Bonazzi et al.), which should be included in the discussion. These papers should also be added to the introduction.
- The limitations of this study have been well identified. It might be worth adding some references here (if space permits; e.g. heterogeneity in OAC: Murugaesu et al., Cancer Discovery 2015; heterogeneity in OAC ctDNA studies: Bonazzi et al., ESMO Open. 2022; frequent copy number alterations in OAC: Frankel et al., Genes Chromosomes Cancer 2014/Killcoyne et al., Nature Medicine 2020/Pasello et al., Modern Pathology 2008 – these are based on a very quick and brief PubMed search, there might be better fitting and more recent publications).
- The discussion acknowledges that EAC and ESCC cases were included and that there were no outcome differences observed here. If the authors would like to keep the combined cohort, I recommend adding this to the results.
